# Image Generation using Continuous Filter Atoms

**Ze Wang**[*]   **Seunghyun Hwang**[*]   **Zichen Miao**   **Qiang Qiu**
Purdue University
{zewang, hwang229, miaoz, qqiu}@purdue.edu

## Abstract

In this paper, we model the subspace of convolutional filters with a neural ordinary differential equation (ODE) to enable gradual changes in generated images. Decomposing convolutional filters over a set of filter atoms allows efficiently modeling and sampling from a subspace of high-dimensional filters. By further modeling filters atoms with a neural ODE, we show both empirically and theoretically that such introduced continuity can be propagated to the generated images, and thus achieves gradually evolved image generation. We support the proposed framework of image generation with continuous filter atoms using various experiments, including image-to-image translation and image generation conditioned on continuous labels. Without auxiliary network components and heavy supervision, the proposed continuous filter atoms allow us to easily manipulate the gradual change of generated images by controlling integration intervals of neural ordinary differential equation. This research sheds the light on using the subspace of network parameters to navigate the diverse appearance of image generation.

## 1   Introduction

Conditional image generation has been widely studied in recent years due to its numerous applications including image segmentation [12, 18], style transfer [57, 20], image inpainting [33, 52], image super-resolution [54, 44], image registration [1], and image synthesis [31]. Despite extensive research and applications in these fields, limited progress has been made on conditional image generation using continuous or closely spaced labels due to the difficulties pointed out in [8], e.g., the absence of real images for some labels. It is shown in [8] that the standard training with empirical risk minimization does not apply well to continuous labels. Moreover, it remains challenging to encourage the generation diversity, especially without heavy supervision, while maintaining output fidelity. Previously, generation diversity is mainly encouraged using explicit regularization terms to convey diversity with additional latent codes [26, 34]. However, explicit regularizations inevitably introduce additional hyperparameters, and sub-optimal hyperparameters can often cause either poor diversity or noticeable sacrifices in terms of generation quality and the correspondence to input conditions.

On the other hand, achieving generation diversity by modeling the deep network parameter space has attracted limited attention [47]. This direction is mainly challenged by the prohibitive cost on both modeling of and sampling from the very high-dimensional space of convolutional filters in modern image generative networks. Motivated by the observation that a convolutional filter can be well approximated by a linear combination of low-dimensional filter atoms, BasisGAN [47] shows in image generation that, the high-dimensional filter space in each layer can be well approximated by a low-dimensional filter subspace, which significantly reduces the cost of both parameter modeling and sampling. In this way, each sampling of the modeled parameter subspace results in one deterministic transformation from the input condition to the desired target domain, and the diverse outputs are achieved by sampling multiple times. More importantly, it is empirically shown in [47] that without

---

[*]Equal contribution.

35th Conference on Neural Information Processing Systems (NeurIPS 2021).

any explicit regularizations, such parameter subspace modeling can work as a plug-and-play module to convert a deterministic conditional image generation network to produce diverse images with appealing stochasticity, without any auxiliary network components or regularization terms.

However, we observe that the filter subspace modeling in [47] suffers from several major limitations: First, this method is sensitive to network configurations of the basis generator, as minor changes to the parameters can highly likely to trigger mode collapse in the parameter distribution, which results in a point estimation to the subspace of parameter and removes diversity in the output images. Furthermore, BasisGAN is incapable of modeling a continuous space, so that gradual changes to the generated parameters cannot be obtained by simply interpolating the latent codes.

In this paper, we adopt a subspace view to convolutional filters by performing atom-coefficients decomposition, as in [35, 47, 46, 48]. Then we further model the filter subspace using a neural ordinary differential equation (ODE), so that we are able to sample from this continuous subspace a series of filter atoms at arbitrarily fine resolution. The low-dimensional filter subspace allows extremely efficient learning and modeling using ODE. We show both empirically and theoretically that, continuous transition in the filter subspace can be propagated naturally to generated images, and thus produce visually appealing images with gradually varying appearance. More importantly, we show that the continuity introduced to the filter subspace and the induced generation smoothness allow continuous manipulation of generated appearance with only discrete samples as supervision, which is a task that is used to be considered intractable by training with standard empirical risk minimization [8]. These appealing properties of the proposed method enable various significant applications through only standard training, without requiring any auxiliary network components, additional regularization terms, or heavy supervision.

We perform various experiments on conditional image generation, with conditions being in the form of images, labels, or both. We list here several example applications that are enabled by the proposed image generation using continuous filter atoms from an ODE atom generator:

- Continuous image synthesis that covers a wide range of gradually varying appearance with high fidelity and accurate correspondence to the input condition.

- Sequential image synthesis with explicitly specified starting and ending points to allow flexible appearance manipulation without heavy supervision.

- Interpolation of generated image appearance at arbitrarily fine resolution.

- An effective approach to generating images conditioned on continuous labels.

## 2   Related Work

**Conditional image generation.** Image generation is widely studied in the machine learning community, from restricted Boltzmann machines [39] to variational autoencoders [21]; in particular variants with conditions [30, 40, 42] show promising results. Empowered by GANs [11], conditional generative adversarial networks (cGANs) [17, 33, 38, 43, 50, 57, 26, 47] have demonstrated superior performance on image generation with the restriction of condition. Recently, conditional image generation with conditions on continuous labels has attracted attention, and new training schemes [8] are proposed to improve continuous conditional image generation. From the perspective of the dynamic of parameter subspace, our method provides a simple yet effective solution.

**Continuous image translation.** Many works have discussed improving the diversity of generated images in image-to-image translation tasks [47, 34, 6, 26, 23, 45, 15, 53, 5]. However, continuous image translation that aims to generate intermediate images between the source and the target images is not yet widely studied. Some previous works [55, 51, 27, 49, 10] have manipulated the attribute vectors either by interpolation or linear transformations to continuously control the latent space. [16, 25] employ multiple generative networks in the middle of the pathway from source to target image to generate intermediate results. [6, 36] disentangle the latent space as content and style codes and interpolate the styles, in order to produce intermediate samples. However, all the aforementioned methods rather need to import auxiliary neural networks or carry extra burdens on model training by introducing additional regularization or supervisions. Moreover, synthesizing intermediate images without relying on interpolation of either domain labels, or disentangled representations, which can discourage the stochasticity of generated images, is still considered a big challenge.

# 3 Continuous Filter Atoms

In this section, we start with a brief review of neural ODEs and atom-coefficient filter decomposition, which serve as key ingredients in the proposed image generation approach. We then present in detail that, with the effectiveness being theoretically justified, how the proposed continuous atom generation using neural ODEs achieves gradual image generation.

## 3.1 Neural ODEs

Neural ordinary differential equations (ODEs) are introduced in [4] as an approach to interpreting the neural network as a system of ODEs, where each ODE represents an underlying dynamic of the hidden elements. Especially, neural ODEs are frequently adopted to enforce continuous dynamics in latent representations, for example, [19, 32] propose to use neural ODEs to smoothly connect the latent space of video frames; [37] adopts ODE-RNNs to model time-continuous hidden dynamics to handle irregularly sampled time-series data.

Neural ODEs model a latent state $\mathbf{z}(t_s)$ as $\mathbf{z}(t_s) = \mathbf{z}(t_0) + \int_{t_0}^{t_s} f(\mathbf{z}(t), t; \boldsymbol{\theta}) \, dt$, where $\frac{d\mathbf{z}(t)}{dt} = f(\mathbf{z}(t), t; \boldsymbol{\theta})$ is modeled as a neural network parametrized by $\boldsymbol{\theta}$. Then, the latent state at an arbitrary point $t_s$ can be obtained as

$$\mathbf{z}(t_s) = \text{ODESolve}(\mathbf{z}(t_0), f, (t_0, t_s), \theta), \tag{1}$$

where $t_0$ and $t_s$ denote the start and the end of the integral interval, respectively. The simplest method for approximating the solutions of the ordinary differential equations with a given initial value is the Euler's method, which is a first-order integrator with a fixed step size. In practice, families of Runge-Kutta methods, e.g., the Midpoint method and fourth-order Runge-Kutta [22], are preferred due to their superior convergence and stability. Note that all the experiments in this paper follow the implementation in [4] for Runge-Kutta of fifth-order of Dormand-Prince-Shampine with adaptive step size [9].

For any given integral interval $(t_i, t_j)$ where $i, j \in \{0...T\}$, ODESolver always outputs a unique solution for the integral of the continuous dynamics, as long as the ordinary equation $f$ is uniformly Lipschitz continuous in $\mathbf{z}$ and $t$ [7]. Parametrizing $f$ as a neural network equipped with Lipschitz continuous non-linear functions will meet the requirements. Thus, in our paper, modeling filter subspace using neural ODEs guarantees continuity as we will theoretically justify later.

## 3.2 Convolutional Filter Decomposition

Directly modeling the space of high-dimensional convolutional filters in an image generation network is practically prohibitive in terms of both computation and parameter scale [47]. Inspired by the observation that a convolutional filter can be well-approximated as a linear combination of filter bases [35], as shown in [47], a subspace view to the convolutional filters can be adopted in image generation network by decomposing each convolutional filter as a linear combination of low-dimensional filter atoms. Specifically, given an $l \times l$-sized convolutional filter $\mathbf{F} \in \mathbb{R}^{c \times c' \times l \times l}$ with $c'$ input and $c$ output channels, respectively, we can decompose the filter over a set of $m$ atoms $\mathbf{D}$ as $\mathbf{F} = \mathbf{A}\mathbf{D}$, where $\mathbf{D} \in \mathbb{R}^{m \times l \times l}$, and $\mathbf{A} \in \mathbb{R}^{c \times c' \times m}$ are the filter subspace coefficients. As we will show later, by modeling the continuity of a low-dimensional filter subspace using neural ordinary differential equations, we achieve efficient training and modeling of gradual appearance changes in conditional image generation.

## 3.3 Continuous Atom Generation for Smooth Appearance Modeling

In this paper, we aim for modeling gradual changes in generated images with respect to continuous input conditions. Instead of seeking new ways of training, we provide a unique view by modeling the transformation from an input condition to the corresponding output with parameters instantiated from an underlying continuous space, which subsequently leads to generated images with gradual changes.

We adopt the aforementioned atom-coefficient decomposition, which permits low computational modeling of filter subspace as also validated in [47]. However, modeling a filter subspace with standard neural networks can still suffer from mode collapse often caused by sub-optimal network configurations, and the continuity is also difficult to be guaranteed by standard training as discussed

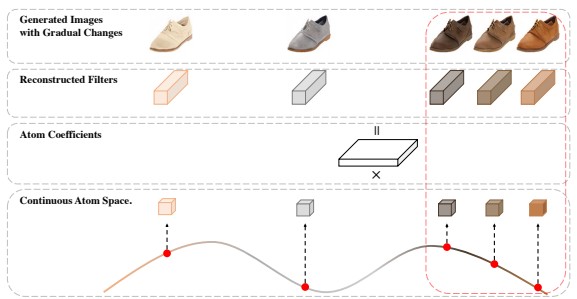

Figure 1: Illustration of the proposed image generation using continuous filter atoms. Note that only a single convolutional layer is shown here for easier viewing. We decompose each convolutional filter as $\mathbf{F} = \mathbf{AD}$, where the coefficients $\mathbf{A} \in \mathbb{R}^{c \times c' \times m}$ remain learned deterministically, and the filter atoms $\mathbf{D} \in \mathbb{R}^{m \times l \times l}$ are sampled from a continuous atom space modeled with neural ODE. As shown in the red box, continuously sampled atoms can produce images with gradual appearance changes.

in [8]. We therefore adopt a neural ODE based filter atom generator with the continuity property discussed in Section 3.1, and show theoretically that such atom continuity can be propagated to the deep network outputs to achieve conditional image generation with gradual changes.

Specifically, we decompose the $i$-th layer filters $\mathbf{F}^i$ into respective atoms $\mathbf{D}^i$ and coefficients $\mathbf{A}^i$, as $\mathbf{F}^i = \mathbf{A}^i \mathbf{D}^i$. Over $k$ layers of filters $\{\mathbf{F}^i\}_1^k$, we jointly model the corresponding atoms $\{\mathbf{D}^i\}_1^k$ using atom generation. We assume that all instantiations of filter atoms reside in a continuous space parametrized as $\boldsymbol{\mathcal{D}}(t)$ where $t$ is a time variable, i.e., the generated atoms at the time point $t$ can be expressed as $\boldsymbol{\mathcal{D}}(t)$. In the limit of a small step $\epsilon$, the local dynamic within this atom space can then be expressed as $\boldsymbol{\mathcal{D}}(t + \epsilon) = \boldsymbol{\mathcal{D}}(t) + \epsilon \cdot \mathrm{d}\boldsymbol{\mathcal{D}}(t)/\mathrm{d}t$. Therefore, given an initial state of atoms within this space denoted as $\boldsymbol{\mathcal{D}}(t_0)$, and modeling the derivative of $\mathrm{d}\boldsymbol{\mathcal{D}}(t)/\mathrm{d}t$ using a neural network $f(\cdot, \cdot; \theta)$:

$$\frac{\mathrm{d}\boldsymbol{\mathcal{D}}(t)}{\mathrm{d}t} = f(\boldsymbol{\mathcal{D}}(t), t; \theta), \tag{2}$$

any samples within this atom space can then be obtained by an integral of the function values of $f$ from $t_0$ to the desired ending point. The described dynamic leads naturally to a neural ordinary differential equation, where the instantiation at a time point $t_s, s > 0$, in this continuous dynamic can be obtained through a continuous atom generator $\mathcal{T}$ by

$$\boldsymbol{\mathcal{D}}(t_s) = \mathcal{T}(t_s; \boldsymbol{\mathcal{D}}(t_0), \theta) = \boldsymbol{\mathcal{D}}(t_0) + \int_{t_0}^{t_s} f(\boldsymbol{\mathcal{D}}(t), t; \theta) dt. \tag{3}$$

Note that the initial state $\boldsymbol{\mathcal{D}}(t_0)$ here are free parameters to be optimized in the training jointly with $\theta$ that parametrizes $f$. Starting from the initial state $\boldsymbol{\mathcal{D}}(t_0)$, an instantiation of filter atoms $\boldsymbol{\mathcal{D}}(t_s)$ is defined to be a unique solution to the ODE initial value problem at some condition $t_s$, by Picard's existence theorem. Consequently, as $t$ varies, different instantiations of atoms are generated from this modeled continuous filter subspace, and our corresponding atom generator equipped with ODESolver learns the underlying continuous dynamics of the filter subspace.

Given the subspace of filters modeled as an underlying continuous dynamic through neural ODE, we prove the Lipschitz-type continuity transducted from atoms to output features in the $i$-th layer. For simplicity, let $c = c' = 1$, and the argument extends. Given an input image $X(u)$ $(u \in \mathcal{U}, \mathcal{U} \subset \mathbb{Z}^2)$, define the local input norm $||x||_{2,N_u} := (\sum_{u' \in N_u} x(u - u')^2)^{1/2}$ and the convolution $\langle x, w \rangle_{N_u} := \sum_{u' \in N_u} x(u - u')w(u')$, where $N_u \subset \mathcal{U}$ is a local Euclidean grid centered at $u$. The $i$-th layer function $\mathcal{F}^i : X^{i-1} \to X^i$ can be expressed as,

$$X^i(u) = \mathcal{F}^i(X^{i-1}, \mathbf{D}^i; \mathbf{A}^i) = \sigma(\sum_{k=1}^{m} \mathbf{A}_k^i \langle X^{i-1}, \mathbf{D}^{i,k} \rangle_{N_u} + b^i), \tag{4}$$

where $\mathbf{D}^{i,k}$ denotes the $k$-th atom in the $i$-th layer, $\mathbf{A}_k^i$ is the corresponded $k$-th coefficient, and $b^i$ denotes the bias at the $i$-th layer.

**Theorem 1.** *Suppose $\mathbf{D}_1^i$ and $\mathbf{D}_2^i$ are two continuously generated atoms, (i.e., $\exists M > 0, ||\mathbf{D}_1^i - \mathbf{D}_2^i||_2 \le M|t_1 - t_2|$), and assume the activation function $\sigma$ is non-expansive which holds for ReLU, then $\mathcal{F}^i$ is continuous in $\mathbf{D}^i$,*

$$||X_1^i - X_2^i||_2 \le (||\mathbf{A}^i||_2 \lambda)\sqrt{|\mathcal{U}|} \cdot ||(\mathbf{D}_1^i - \mathbf{D}_2^i)||_2, \quad with \ \lambda = \sup_{u \in \mathcal{U}} ||X^{i-1}||_{2,N_u}, \tag{5}$$

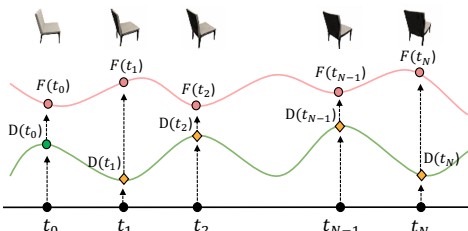

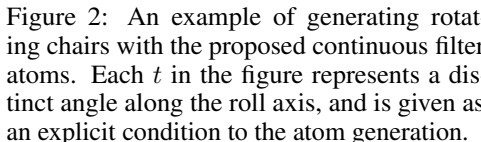

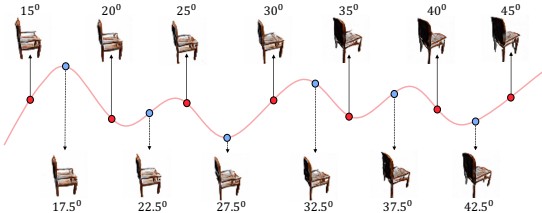

Figure 2: An example of generating rotating chairs with the proposed continuous filter atoms. Each $t$ in the figure represents a distinct angle along the roll axis, and is given as an explicit condition to the atom generation.

Figure 3: Visualization of interpolated samples in the rotating chair example. As the neural ODE-based atom generator learns a smooth trajectory of rotation, image generation at any desired degrees is now allowed. The images generated at the blue dots are interpolation results for unseen rotation angles.

in which $X_1^i = \mathcal{F}^i(X^{i-1}, \mathbf{D}_1^i; \mathbf{A}^i), X_2^i = \mathcal{F}^i(X^{i-1}, \mathbf{D}_2^i; \mathbf{A}^i)$ are outputs correspond to two atoms.

*Proof.* The proof is provided in the Appendix Section A.

Notably, modeling the atom subspace of filters offers significant advantages on efficiency. Under the rule of neural ODE, the dimensionality of $\mathcal{D}(t)$ cannot be varied, meaning the dimension of the input and the output of the ODESolver remain strictly consistent. This additional restriction makes modeling the space of full-size filters practically infeasible due to the computation and memory constraints. Meanwhile, by modeling the filter subspace instead, we significantly reduce the cost of learning the corresponding neural ODE. Typically, atoms across layers introduce a few hundred parameters only and significantly simplify the training and inference of atom generator $\mathcal{T}$.

### 3.4   Image Generation with Continuous Labels

Depending on the specific applications of the proposed continuous atom generation as we will show next, the ending points of the ODE integral can be: 1) explicitly given to strictly control the behavior of continuous dynamics, or 2) stochastically sampled from a prior distribution, e.g., Uniform or Gaussian, to enhance the diversity of the generated images.

**Explicit continuous conditions.**   Compared to the previous conditional GAN methods [28, 29], where label information is vectorized or one-hot encoded, the proposed method permits minimal preprocessing to the continuous labels, which are fed into the continuous atom generator simply in the forms of raw floating numbers preserving the continuity of labels. Moreover, unlike [28] which can only take finite discretized conditions, such as categorical class information, continuous filter atoms do not hold any such restrictions. Therefore, by sampling filter atoms from the subspace of convolutional filters modeled by neural ODE, with the desired ending points of integral specified by the continuous labels, the continuous filter atoms can be evaluated with any values within a valid range of the regression labels. To illustrate this, we present an interpolation example in Figure 3, and show that, with the underlying dynamics of the rotated chair images captured by continuous atom generator trained on limited angels only, the proposed method allows interpolation to the modeled dynamic at arbitrarily fine resolution, and generates rotated chairs at any angels that are not exposed to the model during training.

**Stochastic continuous conditions.**   The stochastic modeling of integration interval permits generated images with diverse appearances, and gradual transitions among the generated images. As we will show in Section 4.1, while the proposed method cannot directly enforce perceptual diverse samples as in [47], the sampled sequences with gradual appearance changes jointly offer decent diversity while maintaining outstanding fidelity and correspondence to the input conditions.

### 3.5   Conditional Image Generation with Continuous Atoms

By simply plugging in the proposed continuous atom generator to off-the-shelf GANs, we show next that variants of conditional generative adversarial models (cGAN) [11, 28] with continuity can

be easily obtained. The training *does not* involve any auxiliary network components or additional regularizations.

***Conditional on images.*** Image generation conditional on images, also known as image-to-image translation [17], aims at generating a target image of a certain domain with high fidelity and desirable diversity [47, 26, 34] while maintaining correspondence w.r.t. an input image condition. We introduce image generation conditional on input images with the output space modeled as a smooth sequence through continuous atom filters, thus sampling the continuous filter atoms allows generating images with gradual appearance changes without losing the correspondence to the conditions. Formally, representing the condition images as $\mathbf{x}$, which are usually samples from an empirical distribution $p(\mathbf{x})$, we aim at modeling a conditional distribution $p(\mathbf{y}|\mathbf{x})$. By plugging in the continuous atom generator $\mathcal{T}(t; \boldsymbol{\mathcal{D}}_0, \theta)$, the conditional distribution is learned to be modeled by generator $\mathbb{G}_{\phi, \theta, \boldsymbol{\mathcal{D}}_0}$, where $\phi$ contains all remaining parameters in the generator, e.g., the coefficients $\mathbf{A}$ in each decomposed convolutional layer. And the input to the discriminator $\mathbb{D}$ is now trained to evaluate the compatibility of a pair of images, one from the source domain $\mathbf{x} \sim p(\mathbf{x})$, the other one comes from either true target distribution $\mathbf{y} \sim p(\mathbf{y}|\mathbf{x})$, or the generated approximation $\mathbb{G}_{\phi, \theta, \boldsymbol{\mathcal{D}}_0}(\mathbf{x}; \mathcal{T}(t; \boldsymbol{\mathcal{D}}_0, \theta))$.

The training can be realized as

$$
\begin{aligned}
\min_{\mathbb{G}} \max_{\mathbb{D}} \mathcal{L}(\mathbb{D}, \mathbb{G}) = & \mathbb{E}_{\mathbf{x} \sim p(\mathbf{x}), \mathbf{y} \sim p(\mathbf{y}|\mathbf{x})}[\log(\mathbb{D}(\mathbf{x}, \mathbf{y}))] + \\
& \mathbb{E}_{\mathbf{x} \sim p(\mathbf{x}), t \sim \mathcal{U}(t_0, t_{-1})}[\log(1 - \mathbb{D}(\mathbf{x}, \mathbb{G}_{\phi, \theta, \boldsymbol{\mathcal{D}}_0}(\mathbf{x}; \mathcal{T}(t; \boldsymbol{\mathcal{D}}_0, \theta))))],
\end{aligned}
\tag{6}
$$

where $\mathcal{U}(t_0, t_{-1})$ denotes uniform distribution with the range specified by $t_0$ and $t_{-1}$.

***Conditional on labels.*** As pointed out in [8], cGANs are good at conditional image generation with discrete labels but can struggle at continuous labels. In our method, without leveraging any customized training schemes as in [8], we show that the continuous image generation with standard empirical risk minimization can be significantly improved by placing an assumption on the gradual appearance change across continuous labels, and trained with continuous atom generation given continuous labels as the inputs. Formally, we define the scalar regression label as $x$ sampled continuously from $x \sim p(x)$, and plug in the continuous atom generator to standard cGAN to model the desired target distribution $p(\mathbf{y}|x)$. The image generation is then achieved as $\mathbb{G}_{\phi, \theta, \boldsymbol{\mathcal{D}}_0}(\mathbf{z}; \mathcal{T}(\alpha x; \boldsymbol{\mathcal{D}}_0, \theta))$, with an input vector $\mathbf{z} \sim \mathcal{N}(0, \mathbf{I})$ carrying randomness as in standard cGANs. Comparing to (6), the input to the continuous atom generator is replaced as $\alpha x$, where $\alpha$ is a predefined scalar that linearly scales the input continuous labels. And we use a standard discriminator with each input consists of an image and a label. The training can be realized as

$$
\begin{aligned}
\min_{\mathbb{G}} \max_{\mathbb{D}} \mathcal{L}(\mathbb{D}, \mathbb{G}) = & \mathbb{E}_{x \sim p(x), \mathbf{y} \sim p(\mathbf{y}|x)}[\log(\mathbb{D}(x, \mathbf{y}))] + \\
& \mathbb{E}_{x \sim p(x), \mathbf{z} \sim \mathcal{N}(0, \mathbf{I})}[\log(1 - \mathbb{D}(x, \mathbb{G}_{\phi, \theta, \boldsymbol{\mathcal{D}}_0}(\mathbf{z}; \mathcal{T}(\alpha x; \boldsymbol{\mathcal{D}}_0, \theta))))].
\end{aligned}
\tag{7}
$$

***Conditional on both images and labels.*** For image-to-image translation tasks, a few works have explored imposing conditions on both images and labels to achieve scalable image translation while introducing diversity and manipulations. Specifically, [45] and [5] impose additional domain attributes to navigate the image translation. However, the domain attributes they condition on are all categorical labels, and none of those methods can adopt continuous or raw format (e.g, float numbers) labels to capture the gradual changes in the desired target image space. Our method permits using the continuous labels fed into the atom generator to explicitly control sampling in the continuous filter subspace, while maintaining consistent input image conditions. Therefore, the proposed method is able to achieve image generation with gradual appearance changes w.r.t. the input continuous label conditions, and preserve strong correspondence to the source image. To the best of our knowledge, our method is the first work to present gradually changing image-to-image translation results conditioned on both images and continuous labels. Formally, having both $\mathbf{x} \sim p(\mathbf{x})$ and $x \sim p(x)$, which refer to the input image and the scalar label, respectively, we employ our continuous atom generator, $\mathcal{T}(\alpha x; \boldsymbol{\mathcal{D}}_0, \theta)$, to get the desired target distribution $p(\mathbf{y}|x, \mathbf{x})$. We then formulate our image generator as $\mathbb{G}_{\phi, \theta, \boldsymbol{\mathcal{D}}_0}(\mathbf{x}; \mathbf{z}; \mathcal{T}(\alpha x; \boldsymbol{\mathcal{D}}_0, \theta))$. The training can be realized as

$$
\begin{aligned}
\min_{\mathbb{G}} \max_{\mathbb{D}} \mathcal{L}(\mathbb{D}, \mathbb{G}) = & \mathbb{E}_{\mathbf{x} \sim p(\mathbf{x}), \mathbf{y} \sim p(\mathbf{y}|\mathbf{x}, x)}[\log(\mathbb{D}(\mathbf{x}, \mathbf{y}))] + \\
& \mathbb{E}_{\mathbf{x} \sim p(\mathbf{x}), x \sim p(x)}[\log(1 - \mathbb{D}(\mathbf{x}, \mathbb{G}_{\phi, \theta, \boldsymbol{\mathcal{D}}_0}(\mathbf{x}; \mathcal{T}(\alpha x; \boldsymbol{\mathcal{D}}_0, \theta))))].
\end{aligned}
\tag{8}
$$

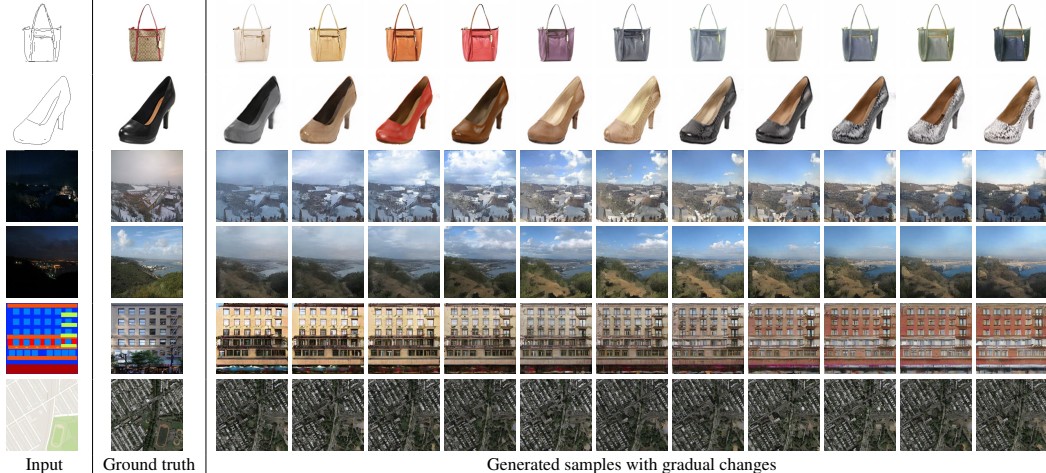

| Input | Ground truth | Generated samples with gradual changes |
|---|---|---|

Figure 4: Continuous image-to-image translation. The network is trained without any auxiliary loss functions or regularizations. From top to bottom, the image to image translation tasks are: edges → handbags, edges → shoes, nights → days, labels → facades, and maps → satellite.

Table 1: Quantitative results on image generation conditional on images only.

| Methods | Labels→Facedee | | Map→Satellite | | Night→Day | | Edge→Shoe | | Edge→Handbag | |
|---|---|---|---|---|---|---|---|---|---|---|
| | LPIPS ↑ | FID ↓ | LPIPS ↑ | FID ↓ | LPIPS ↑ | FID ↓ | LPIPS ↑ | FID ↓ | LPIPS ↑ | FID ↓ |
| BicycleGAN | 0.1413 | 98.85 | 0.1150 | 145.78 | 0.103 | 120.63 | 0.139 | 72.49 | 0.184 | 96.28 |
| MSGAN | 0.1894 | 92.84 | 0.2189 | 152.43 | 0.176 | 107.90 | 0.167 | 60.28 | 0.228 | 89.96 |
| BasisGAN | 0.2648 | 88.70 | **0.2417** | **35.54** | 0.184 | **102.56** | **0.242** | 64.17 | **0.350** | 88.76 |
| Ours | **0.2712** | **87.75** | 0.1803 | 98.72 | **0.264** | 106.95 | 0.237 | **57.29** | 0.276 | **38.07** |

## 4 Experiments

In this section, we present results and comparisons in two experimental settings. We start with continuous image-to-image translation tasks where we show that by directly plugging in the proposed continuous atom generator, we can either condition the generated images solely on images, or on both images and labels, where each of the cases refers to (6) and (8), respectively. In both cases, we quantitatively and qualitatively show that our method achieves continuous image generation with high fidelity and diversity. Then, we proceed to image generation tasks conditional on continuous labels following (7). We compare the proposed method with [8] and [28] on the face generation experiments. Implementation details are in Appendix Section B. More qualitative results are in Appendix Section C.

### 4.1 Continuous Image-to-image Translation

The introduced continuous atom generation can be naturally applied to the image-to-image translation experiments [47] as a simple plug-and-play module. Specifically, in image-to-image translation tasks with paired samples, we adopt a ResNet [13] based generator network following the implementation of Pix2Pix [17]. All the convolutional filters of the ResNet blocks are decomposed into atom-coefficients where an individual atom of each block is generated continuously with neural ODE. Note that a single neural ODE is utilized throughout the entire network, enforcing the network to be governed by a single continuous dynamic. These settings are also applied in unpaired image-to-image translation tasks, where we replace our baseline model from the Pix2Pix to ResNet-based CycleGAN [57].

**Continuous image-to-image translation conditional on images.** We present results on various datasets including edges → handbags, edges → shoes, maps → satellite, night → days, and labels → facades. Compared to previous methods working on the same datasets, our method does not focus on generating images with strong diversity. Instead, we show that by plugging in the introduced continuous filter atoms, images with smooth intermediate transitions can be obtained by directly

Table 2: Quantitative results on image generation conditional on both images and labels under the configuration of image-to-image translation tasks with paired inputs. Diversity and fidelity are measured using LPIPS and FID, respectively. As Pix2Pix is a deterministic model, LPIPS scores for Pix2Pix are all marked as '-', and all the scores resulted in zero.

| Datasets | RC49 | | RCA20 | | RA20 | | RL20 | | RT20 | |
|---|---|---|---|---|---|---|---|---|---|---|
| Methods | Pix2Pix | Ours | Pix2Pix | Ours | Pix2Pix | Ours | Pix2Pix | Ours | Pix2Pix | Ours |
| LPIPS ↑ | - | 0.1127 | - | 0.0931 | - | 0.068 | - | 0.1089 | - | 0.1213 |
| FID ↓ | 251.49 | **195.01** | 218.88 | **85.06** | 165.01 | **73.82** | 227.04 | **124.83** | 206.49 | **70.92** |

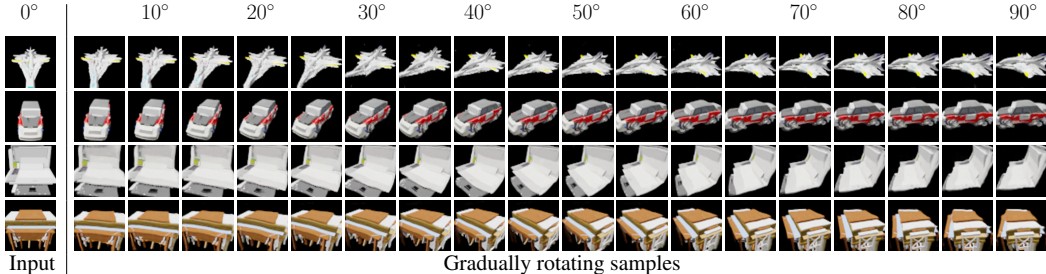

Figure 5: Continuous image-to-image translation conditional on both images and labels with paired samples. From top to bottom, the trained datasets are: RA-20, RCA-20, RL-20, and RT-20. The generated images of all datasets rotate from a degree 0.1 to 89.9 at the interval of 5 degree, along the yaw axis. All models are trained with Pix2Pix by plugging in our continuous atom generator.

training the network without any additional supervision and regularizations. As shown qualitatively in Figure 4 and quantitatively in Table 1, these smooth transitions in the generated appearance can naturally lead to satisfying diversity without compromising the quality of the generated images and the correspondence to the input conditions.

**Continuous image-to-image translation conditional on regression labels.** We present results of image-to-image translation tasks conditional on regression labels on a variety of datasets and experimental settings. Following (8), the generated images are conditional on explicitly given regression labels, while keeping correspondence to the input image. We start with experiments on image-to-image translation on paired samples, and then proceed to the unpaired samples.

*Paired samples.* In this experiment, we plug in our continuous filter atoms to the Pix2Pix [17] model and conduct continuous conditional image-to-image translation tasks on five different synthetic datasets, RC-49 [8], RA-20, RCA-20, RT-20, RL-20, where objects in each dataset are rendered to rotate between 0° and 360° with 0.1° interval. Since there are very few datasets of 3-D objects with continuous labels, we created new datasets RA-20, RCA-20, RT-20, RL-20, by rendering 'Airplanes', 'Cars', 'Tables', 'Laptops' categories of ShapeNet, respectively, following the manuals of RC-49. We add further descriptions of datasets in the Appendix Section B.1. During training, we only use samples with rotation degrees below 90°, and fix the 0° image as the source image with varying target images. We adopt the gap of rotation degree between the source and the target image as an external condition for atom generation. We show in the Figure 5, that the generated images rotate smoothly according to the given rotation degrees. Especially for the results of RC-49, although we have organized the training data in 5° interval, it is noteworthy that our method generated interpolation results at the interval of 2.5 degree in the Figure 3. Table 2 also demonstrates that the generated images of our proposed method hold a higher level of fidelity than the baseline.

*Unpaired samples.* For translation tasks on unpaired datasets, we present results on UTKFace [56], SteeringAngle, cells-200, and Waymo [41] dataset. Other than the Waymo dataset [41], all the datasets are composed of images with continuous labels, e.g., the age for UTKFace, the angle of the steering wheel of a car for the SteeringAngle, and the number of cells for Cells-200. Waymo [41] dataset has four different time labels for each image, which are dawn, daytime, dusk, and night. We add more detailed descriptions on the datasets in the Appendix Section B.1 with additional results of aging face experiments on the CACD2000 [3] at Appendix Section B.2. In the unpaired image-to-image translation tasks, using continuous labels can lead to confusion in training, as the

Table 3: Quantitative results on conditional image generation when image and label were both given as input conditions under the configuration of unpaired image-to-image translation task. We present results on UTKFace [55], CACD2000[3], SteeringAngle, and Cell200. Diversity and fidelity are measured using LPIPS and FID, respectively. The LPIPS scores for the CycleGAN are marked as '-', as the all LPIPS scores resulted in zero, with no meaningful diversity observed.

| Datasets | Cell200 | | SteeringAngle | | UTKFace | | CACD2000 | |
|---|---|---|---|---|---|---|---|---|
| Methods | CycleGAN | Ours | CycleGAN | Ours | CycleGAN | Ours | CycleGAN | Ours |
| LPIPS ↑ | - | 0.1355 | - | 0.0544 | - | 0.0052 | - | 0.0018 |
| FID ↓ | 30.44 | **22.22** | 73.03 | **52.39** | 47.24 | **41.55** | 58.43 | **57.15** |

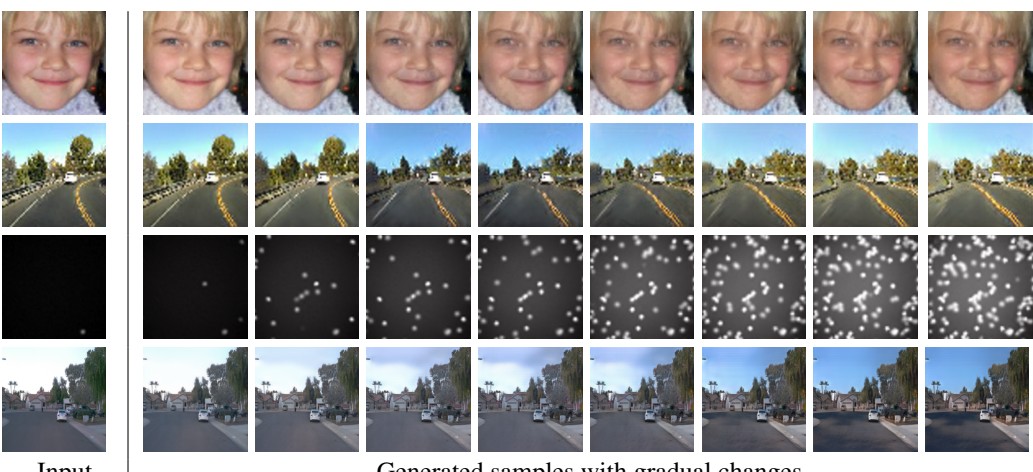

Input | Generated samples with gradual changes

Figure 6: Continuous image-to-image translation conditional on both image and labels with unpaired samples. From top to bottom, the datasets are: UTKFace (young → old), SteeringAngle (steering angle at -80° → steering angle at 80°), Cell200 (one cell → two-hundred cells), and Waymo (dawn → day). All models are trained with CycleGAN by plugging in our continuous atom generator.

tendency of continuous labels can be violated within unpaired samples. For instance, in the human face translation experiment, due to different levels of aging among people, it is more natural to have the age group as a class, rather than treating each age as a class. Therefore, the target domain is divided into a specific interval of labels, so that each sub-group of the target domain can be assigned distinct labels instead. The distinct labels are then directly used as a condition for modeling the continuous subspace of atoms as in (8). Despite the use of distinct labels, we highlight that it is still possible to capture continuous variation between the generated images via interpolation in the continuous filter subspace. During the training, images with certain conditions,.e.g, teenagers for the UTKFace, -65° to -80° for SteeringAngle, one number of cell for Cells-200, and dawn time for Waymo, are fixed as a source domain input. Figure 6 shows the gradual changing between the condition of the target domain. While keeping a certain amount of consistency within the input image, smooth transitions between generated images are observed. In particular, the result of the Waymo dataset at the fourth row of Figure 6 well captures progressive changes over time in the generated images from dawn to daytime, through interpolation in filter subspace. Table 3 also highlights our proposed method's superior fidelity over the baseline model. Additional training details are presented in the Appendix Section B.

## 4.2 Conditional Image Generation with Regression Labels

We present face image generation with the condition only on a regression label specifying the age. The training objective follows (7) strictly. We construct a conditional image generation network by replacing the Adaptive Instance Normalization (AdaIN) [14] layers with standard batch normalization layers, and replace convolutional filters with the proposed continuous atom generation networks. As shown in the qualitative results in Figure 7, by fixing the input latent code **z**, and changing only

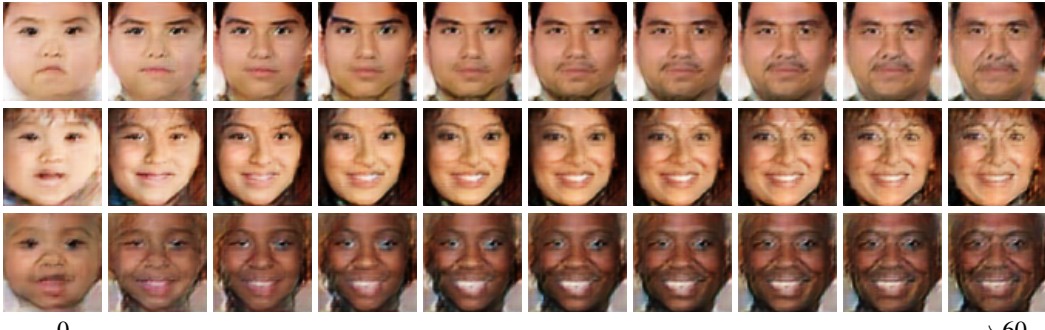

0 ——————————————————————————————→ 60

Figure 7: Continuous image generation conditional on regression labels. From left to right, each column are images generated with the same atoms sampled from the continuous atom space and model the appearance of face at a particular age from 0 to 60. By fixing **z** in (7), each row shows the gradual appearance changes w.r.t. continuously sampled atoms.

Table 4: Quantitative results on conditional image generation when the label is given as an input condition. We present results on UTKFace [56] with age as regression labels.

| Dataset | Methods | Intra-FID ↓ | NIQE ↓ | Diversity ↑ | Label Score ↓ |
|---------|---------|-------------|--------|-------------|---------------|
| UTKFace | cGAN | 4.516 | 2.315 | 0.254 | 11.087 |
| | CcGAN | 0.425 | 1.725 | 1.298 | 7.452 |
| | Ours | 0.432 | 1.749 | 1.321 | 7.399 |

the condition scalar $x$, we are able to generate sequences of images with gradual age changes. And switching **z** allows changing the identity of the face sequence. Comparable quantitative results in Table 4 with CcGAN [8] further demonstrate the effectiveness.

## 5 Conclusion

In this paper, we presented both theoretically and empirically that, continuous dynamics of convolutional filters can be effectively modeled in the filter subspace by neural ordinary differential equation, to subsequently achieve conditional image generation with gradual appearance changes. The introduced continuous filter atom generation enables continuous image generation conditional on images, labels, or both. We demonstrated its effectiveness using both superior quantitative and qualitative results.

## 6 Acknowledgements

This work was supported by the DARPA TAMI program under No. HR00112190038.

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
