# Appendix

## A  Proof of Theorem 2

Recall the operation in the $i$-th layer, i.e., the function $\mathcal{F}^i$, can be expressed as,

$$X^i(u) = \mathcal{F}^i(X^{i-1}, \mathbf{D}^i; \mathbf{A}^i) = \sigma(\sum_{k=1}^{m} \mathbf{A}_k^i \langle X^{i-1}, \mathbf{D}^{i,k} \rangle_{N_u} + b^i), \tag{1}$$

**Theorem 2.** *Suppose $\mathbf{D}_1^i$ and $\mathbf{D}_2^i$ are two continuously generated atoms, (i.e., $\exists M > 0, ||\mathbf{D}_1^i - \mathbf{D}_2^i||_2 \leq M|t_1 - t_2|$), and assume the activation function $\sigma$ is non-expansive which holds for ReLU, then the output $\mathcal{F}^i$ is continuous in $\mathbf{D}^i$,*

$$||X_1^i - X_2^i||_2 \leq (||\mathbf{A}^i||_2 \lambda) \sqrt{|\mathcal{U}|} \cdot ||(\mathbf{D}_1^i - \mathbf{D}_2^i)||_2, \quad \text{with } \lambda = \sup_{u \in \mathcal{U}} ||X^{i-1}||_{2,N_u}, \tag{2}$$

*Proof.* Since $\sigma$ is non-expansive, $\forall u$,

$$|X_1^i(u) - X_2^i(u)| \leq |\sum_{k=1}^{m} \mathbf{A}_k \langle X^{i-1}, \mathbf{D}_1^{i,k} \rangle_{N_u} - \sum_{k=1}^{m} \mathbf{A}_k \langle X^{i-1}, \mathbf{D}_2^{i,k} \rangle_{N_u}|$$
$$\leq ||\mathbf{A}||_2 (\sum_{k=1}^{m} |\langle X^{i-1}, (\mathbf{D}_1^{i,k} - \mathbf{D}_2^{i,k}) \rangle_{N_u}|^2)^{1/2}. \tag{3}$$

By Cauchy-Schwarz inequality,

$$|\langle X^{i-1}, (\mathbf{D}_1^{i,k} - \mathbf{D}_2^{i,k}) \rangle_{N_u}| \leq ||X^{i-1}||_{2,N_u} \cdot ||(\mathbf{D}_1^{i,k} - \mathbf{D}_2^{i,k})||_{2,N_u}$$
$$\leq \lambda \cdot ||(\mathbf{D}_1^{i,k} - \mathbf{D}_2^{i,k})||_2 \tag{4}$$

we have that

$$\sum_{u \in \mathcal{S}} |X_1^i(u) - X_2^i(u)|^2 \leq ||\mathbf{A}||_2^2 \sum_u \sum_{k=1}^{m} |\langle X^{i-1}, (\mathbf{D}_1^{i,k} - \mathbf{D}_2^{i,k}) \rangle_{N_u}|^2$$
$$\leq ||\mathbf{A}||_2^2 \sum_u \sum_{k=1}^{m} ||X^{i-1}||_{2,N_u}^2 \cdot ||(\mathbf{D}_1^{i,k} - \mathbf{D}_2^{i,k})||_{2,N_u}^2 \tag{5}$$
$$\leq (||\mathbf{A}||_2 \lambda)^2 \sum_{u,k} ||(\mathbf{D}_1^{i,k} - \mathbf{D}_2^{i,k})||_2^2$$

and observe that

$$\sum_{u,k} ||(\mathbf{D}_1^{i,k} - \mathbf{D}_2^{i,k})||_2^2 = \sum_{u \in \mathcal{U}} \sum_{k=1}^{m} ||(\mathbf{D}_1^{i,k} - \mathbf{D}_2^{i,k})||_2^2 = |\mathcal{U}| \cdot ||(\mathbf{D}_1^i - \mathbf{D}_2^i)||_2^2, \tag{6}$$

where $|\mathcal{U}|$ is the area of the domain of $X^{i-1}$. Then Eq. 5 continues as

$$\leq (||\mathbf{A}||_2 \lambda)^2 |\mathcal{U}| \cdot ||(\mathbf{D}_1^i - \mathbf{D}_2^i)||_2^2, \tag{7}$$

which proves that $||X_1^i - X_2^i||_2 \leq (||\mathbf{A}||_2 \lambda) \sqrt{|\mathcal{U}|} \cdot ||(\mathbf{D}_1^i - \mathbf{D}_2^i)||_2$ as claimed.

$\square$

## B  Experimental Detail

### B.1  Datasets

We introduce additional descriptions on the datasets with regression labels.

**RC-49.**  RC-49 is first presented in [8], by rendering 3-D chair objects from the 'Chairs' category of the ShapeNet [2]. Objects are originally rendered to rotate from $0°$ to $360°$ at $0.1°$ interval along the roll axis, using Blender v.2.79[1]. In total, the dataset has 49 different shapes of chairs, where we used randomly selected 44 shapes of chairs in training, and the rest for the testing. Note that only chairs at angle between $0°$ and $90°$ are used at both training and testing phases. All images are resized to 256x256 in both training and testing times.

---

[1] https://www.blender.org/download/releases/2-79/

**RCA-20, RT-20, RL-20, RA-20.**    Following the implementation details of RC-49 dataset, we created our own benchmark datasets using the 'Cars', 'Tables', 'Laptops', and 'Airplanes' categories of the ShapeNet [2], and named them RCA-20, RT-20, RL-20, RA-20, respectively. All the datasets are comprised of 3-D objects rotating from $0°$ to $90°$ at $0.1°$ interval along the yaw axis, and Blender v.2.79[1] is used for 3-D rendering. Each dataset contains 20 different shapes of their objects, where randomly selected 18 shapes of each object are used in training, and the rest for the testing. All images are resized to 256x256 in both training and testing times.

**SteeringAngle.**    We used the preprocessed version of SteeringAngle dataset, provided by [8][2]. Originally, the SteeringAngle dataset is a subset of autonomous driving dataset[3], consisted of images taken by a camera mounted on a car, where each image has an angle of the steering wheel rotation of the car as a label. The preprocessed version provided by [8] only contains images at steering angle ranging from $-80°$ to $80°$. When training, we equally divided the entire dataset into 8 different sub-groups with equal angular intervals, e.g., 20 degrees. The distinct labels of each sub group are then normalized between $[0, 1]$ and used as a condition for training different filter atoms. Note that when training CycleGAN, only the first sub-group is adopted as a source domain, and the rest of the groups are all blended together as a target domain. All images are resized to 256x256 in both training and testing times.

**Cell-200.**    We used the preprocessed version of Cell-200 dataset, provided by [8][2]. Cell-200 dataset [24] is a synthetic image dataset that emulates the colonies of bacterial cells in a microscopic view, composed of cell population ranging from 1 to 200. For each population, we used 100 different cell images for training and testing. The number of cells in each image is first normalized between $[0, 1]$ and directly adopted as a condition for modeling the filter atoms. Note that when training CycleGAN, only the first sub-group, that contains one cell in an image is adopted as a source domain, and the rest of the groups are all blended together as a target domain. All images are resized to 256x256 in both training and testing times.

**Waymo**    We used the recent Waymo open dataset [41] for presenting gradual transition of scenes between different timelapse. All images are extracted directly from the video frames, labeled as one of {Day, Dawn, Dusk, Night} time-zones. We resized images into size of 256x256 and converted them into RGB format at both training and testing times. We split images into source {Dusk, Dawn} and target {Day, Night}. The distinct labels of each target group are then normalized between $[0, 1]$, and used as a condition for training different filter atoms. To be more specific, we used 0.0 time label for {Dusk, Dawn}, 0.5 for {Day}, and 1.0 for {Night}.

**UTKFace.**    We used the preprocessed version of UTKFace [56] dataset, provided by [8][2]. The UTKFace dataset is a dataset of RGB images of human faces labeled by age. The preprocessed version contains images of age ranging from 0 to 60, resized to size 64x64. Note that all images are resized again into 256x256 in both training and testing times. We excluded human face images under 10 years old, and the rest of the dataset is divided into six different subgroups, with equal intervals with respect to the age of each image, e.g., 10 years. The distinct labels of each sub group are then normalized between $[0, 1]$ and used as a condition for training different filter atoms. When training CycleGAN, we assigned images with ages ranging from $[10, 30]$ to source domain, and the remaining images to target domain.

**CACD2000.**    We used CACD2000 [3] dataset for demonstrating additional experimental results on our continuous filter atoms. CACD2000 dataset is one of the most popular benchmark dataset for face aging, containing 163,446 face images of 2,000 celebrities collected via Google Image Search. We used Face++ API [4] to crop and save only the face region and center-aligned all images after resizing them into size of 400x400. We only used images of age between 20 to 70. Images are divided into six different subgroups, with equal intervals with respect to the age of each image, e.g., 10 years. The distinct labels of each sub group are then normalized between $[0, 1]$ and used as a condition for training different filter atoms. When training CycleGAN, we assigned images with age ranging from $[20, 30]$ to source domain, and the remaining images to target domain.

## B.2   Evaluation

We use the following evaluation metrics for the comparison against state-of-the-art methods:

**LPIPS.**    The diversity of generated images are measured using LPIPS [26]. LPIPS computes the distance of images in the feature space. Generated images with higher diversity give higher LPIPS scores, which are more favourable in conditional image generation.

---

[2] https://github.com/UBCDingXin/improved_CcGAN.
[3] https://github.com/SullyChen/driving-datasets.
[4] https://www.faceplusplus.com.

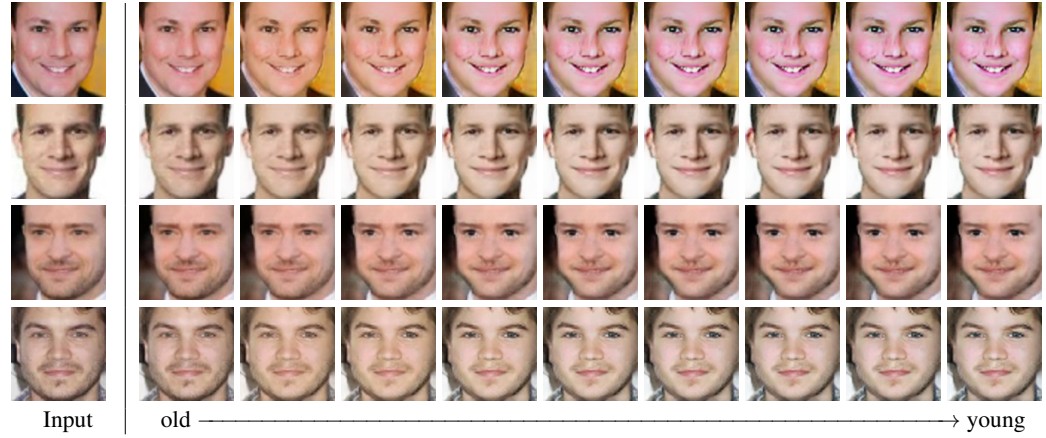

Input | old ⟶ young

Figure A: Additional results on the UTKFace dataset in a reverse direction, from 'old' to 'young', trained with CycleGAN and continuous filter atoms.

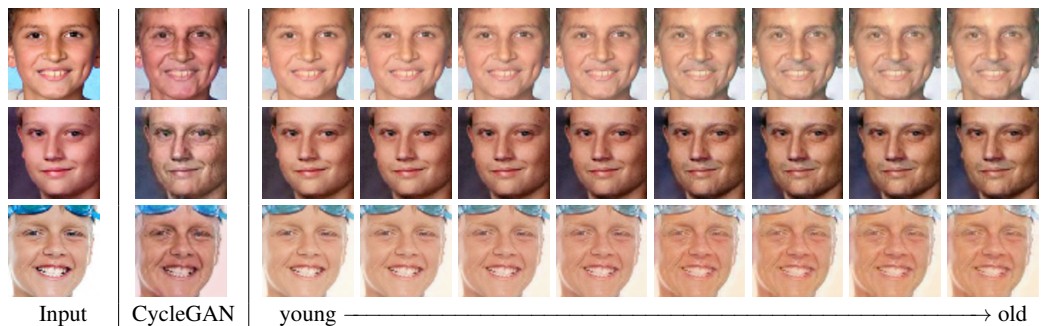

Input | CycleGAN | young ⟶ old

Figure B: Additional qualitative results of human face translation experiments on UTKFace [56] dataset, with comparison to CycleGAN.

**FID.** FID is used to measure the fidelity of the generated images. It computes the distance between the distribution of the generated images and the true images. Since the entire GAN family is to faithfully model true data distribution parametrically, lower FID is favourable in our case since it reflects a closer fit to the desired distribution.

**Intra-FID.** Intra-FID in adopted in Section 4.2, for the evaluation of image generation conditioned on both images and labels. Following [8], the number for Intra-FID is obtained by computing and averaging the FID score at each of the evaluation ages.

**Label score.** Label score is obtained by pre-training a ResNet-34 model for the continuous label regression, and compute the mean absolute error between the predicted and the assigned age of the generated images.

**Diversity.** Diversity score (higher the better) is obtained by pre-training a identity a ResNet-34 classification network, and the results is evaluated as the entropy of the predicted categories of the generated images.

**NIQE** NIQE (lower the better) is used to evaluate the visual quality of fake images with the real images as the reference. We use the MATLAB code provided by [8] for the evaluation.

## C  Qualitative Results

In this section, we present additional qualitative results of experiments introduced in the Section 4.

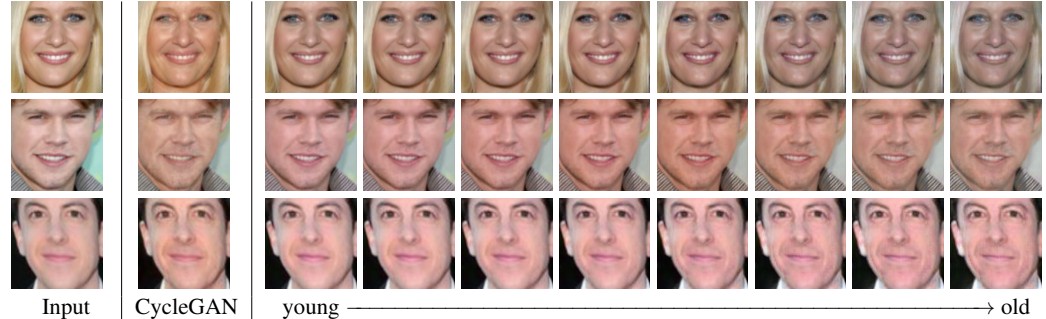

Input | CycleGAN | young ⟶ old

Figure C: Additional qualitative results of human face translation experiments on the CACD2000 [3] dataset with comparison to CycleGAN.

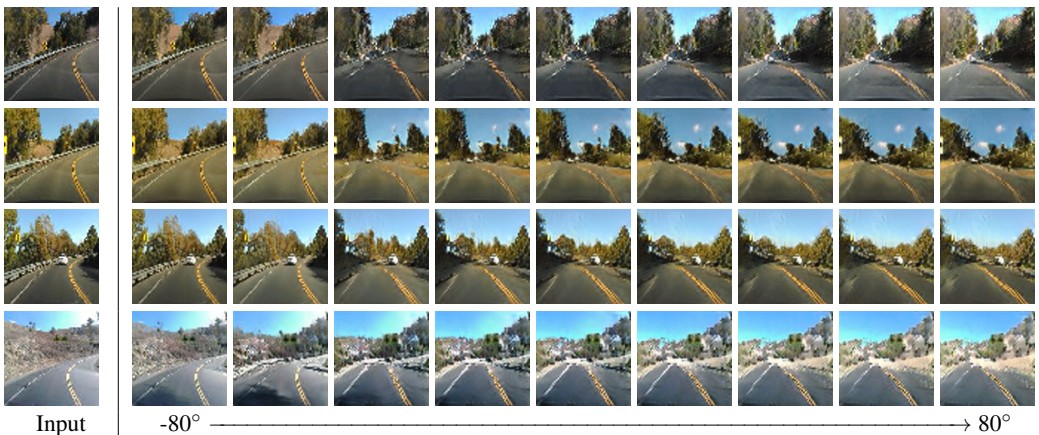

Input | -80° ⟶ 80°

Figure D: Additional results on the SteeringAngle dataset, trained with CycleGAN and continuous filter atoms.

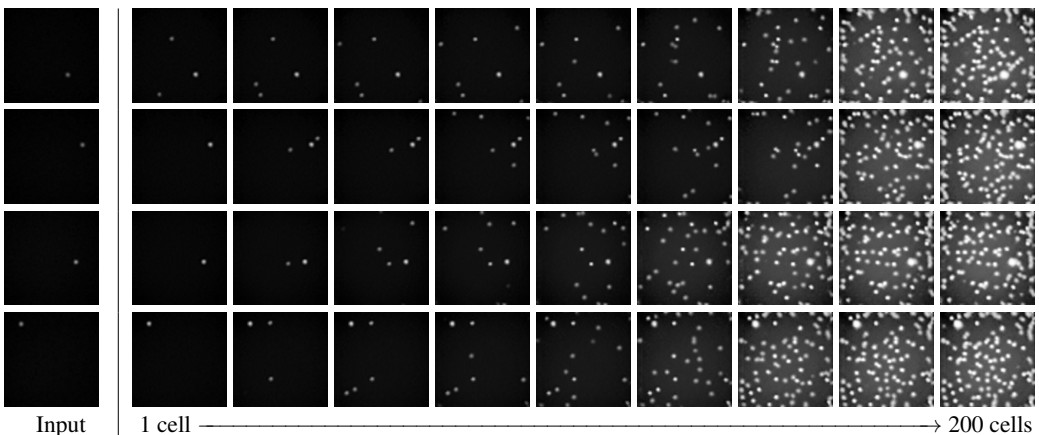

Input | 1 cell ⟶ 200 cells

Figure E: Additional results on the Cells-200 dataset, trained with CycleGAN and continuous filter atoms.

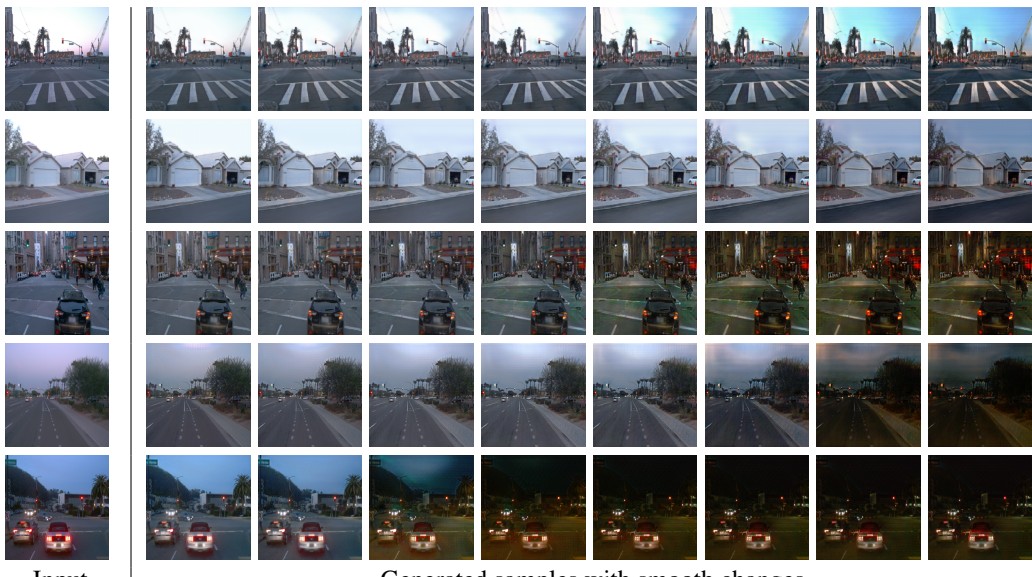

| Input | Generated samples with smooth changes |
|---|---|

Figure F: Additional qualitative results on the Waymo [41] dataset. The top two row shows gradual transition from Dawn to Day, the next two rows show gradual transition from Day to Dusk, and the last two rows show gradual transition from Dusk to Night

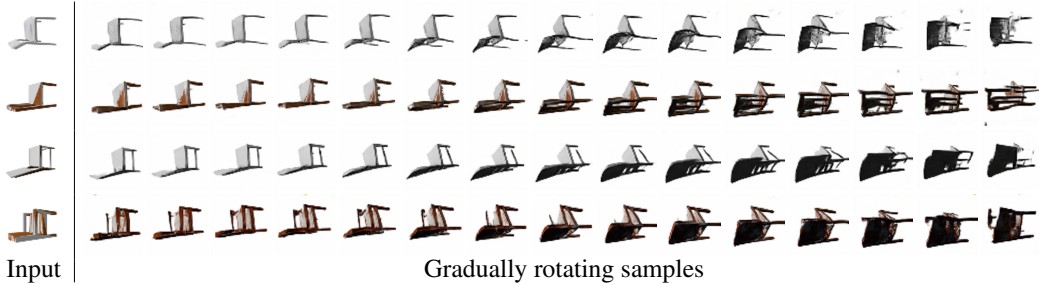

| Input | Gradually rotating samples |
|---|---|

Figure G: Additional results on the RC-49 dataset, trained with Pix2Pix and continuous filter atoms. The model is trained with chairs rotating at $0.1°$ interval.

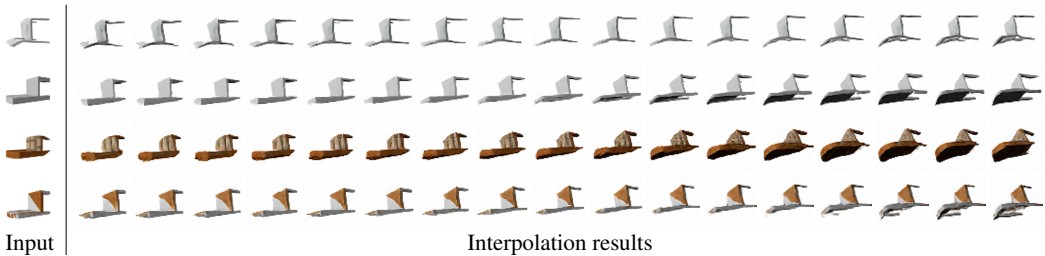

| Input | Interpolation results |
|---|---|

Figure H: Additional interpolation results on the RC-49 dataset, trained with Pix2Pix and continuous filter atoms. Generated chairs rotate from $0°$ to $45°$ at $2.5°$ interval. Note that the model is trained with chairs rotating at $5°$ interval.

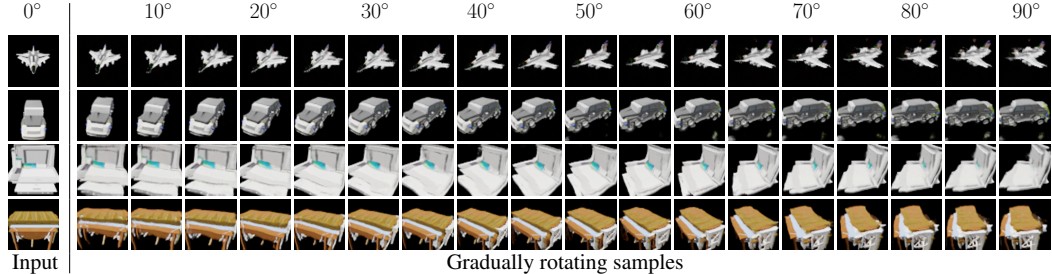

| 0° | 10° | 20° | 30° | 40° | 50° | 60° | 70° | 80° | 90° |

Input | Gradually rotating samples

Figure I: Additional results on the 3-D object datasets, trained with Pix2Pix and continuous filter atoms. From top to bottom: RA-20, RCA-20, RL-20, and RT-20.

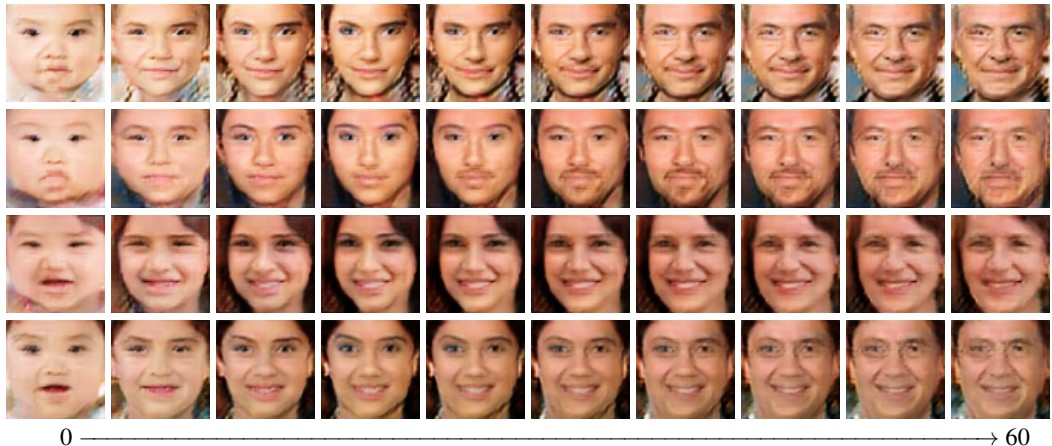

0 ———————————————————————————————→ 60

Figure J: Continuous image generation conditional on regression labels. From left to right, each column are images generated with the same atoms sampled from the continuous atom space and model the appearance of face at a particular age from 0 to 60. By fixing $\mathbf{z}$ in (7), each row shows the gradual appearance changes w.r.t. continuously sampled atoms.

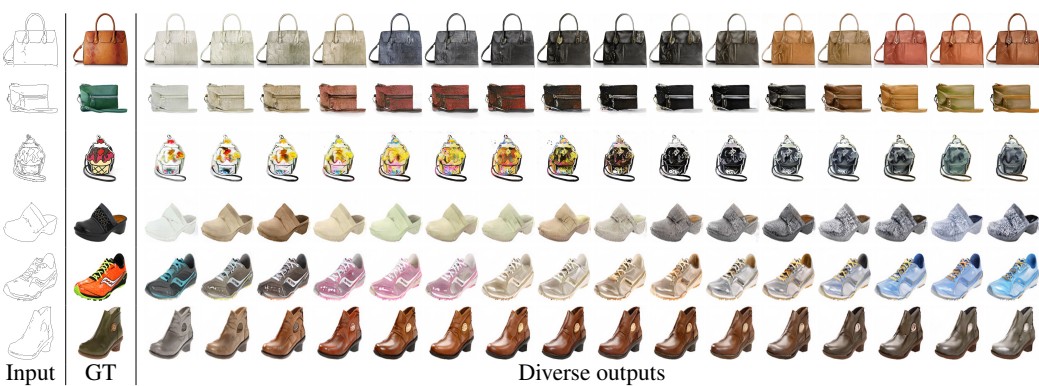

Input | GT | Diverse outputs

Figure K: Continuous image-to-image translation. The network is trained without any auxiliary loss functions or regularizations.