# OpenReview forum: "Image Generation using Continuous Filter Atoms"
_NeurIPS.cc/2021/Conference — NeurIPS 2021 Spotlight_

### Official Review · Reviewer_FG2b · 2021-07-03

**Rating:** 6
**Confidence:** 4

**Summary:**

The presented paper deals with the problem of diversity in produced samples of image generation models. The proposed solution employs the recent framework of decomposing convolutional filters over a set of learned filter atoms. To make the results diverse, one multiplier of such decomposition is modeled with neural ODEs, thus allowing for continuous changes in the outputs of the generator (the related theorem was proved). Importantly, the solution may be integrated into many architectures as a drop-in replacement for standard convolutions.

The authors have conducted experiments on image-to-image translation and image generation with continuous labels. The reported results demonstrate that the ODE-driven filter decomposition leads to meaningful results.

**Limitations And Societal Impact:**

I have no suggestions.

**Main Review:**

1. Strengths.

    a. Neural ODE is a celebrated way of modeling time-continuous dynamics and it looks like a reasonable tool for generative models with non-discrete labels. Also, the framework of Filter Atoms is an interesting alternative for other common conditioning methods like AdaIN, weight (de)modulation, etc. Therefore, the motivation for this paper is pretty clear.

    b. The paper is well-written, all the concepts are explained in a self-contained manner.

2. Weaknesses.

    a. First of all, I have some concerns about experiments from the section "img2img translation conditional on images" (edges -> handbags, etc., line 269). The naturalness of learning the mapping between a one-dimensional time variable and color/satellite style/etc is at least questionable to me. The approach of BasisGAN that models such styles as samples from a multivariate Gaussian seems more reasonable, and this method is well-established in img2img literature (e.g., see [1, 2, 3]). This is possibly why BasisGAN mostly outperforms the authors' model in this task (Table 1). Unfortunately, the authors do not show qualitative results for this problem other than edges -> handbags.

    b. The results on UTKFace and CACD2000, presented in the appendix, look not very convincing to me. The presented approach seems to perform not that well on human faces translation. As Table 3 claims the presented approach outperforms CycleGAN, it would be interesting to demonstrate also the CycleGAN outputs to compare the samples qualitatively.

    c. Why do the authors need to use discrete "distinct labels" (line 304). What exactly makes truly continuous labels "intractable" for the ODE-based approach?

    d. A lot of recent papers [4, 5] have shown the efficiency of positional embedding as a tool for feeding continuous values into the neural network. I think this could be a much simpler baseline aiming to check the usefulness of ODE for continuous conditioning: one may positionally encode the time variable with random or learned frequencies to a vector, then process it with some multi-layer perceptron and obtain a conditioning vector for AdaIN or weight (de)modulation [6]. Such a baseline may be used for experiments described at lines 278 and 312.

    e. The paper lacks hi-res experiments, unlike BasisGAN, which compared itself with the pix2pixHD approach for segmentation -> cityscapes problem. Is the ODE-based approach scalable? How is inference time affected with ODE solver being involved?

3. Suggestions.

    a. The recent CoMoGAN [7] paper also deals with the conditioning on a continuous one-dimensional time variable to model daytime light changes (among other tasks). Although this paper had not been released by the time of the NeurIPS deadline, I think that comparison with CoMoGAN would make the submission stronger.

4. Rating (pre-rebuttal)

    With all the pros and cons listed above, I tend to rate this paper below the threshold (score = 5). To my mind, more evidence to use exactly an ODE-based solution is required. Some of the results look not impressive making the quality of the model questionable. I ask the authors to address the concerns in the rebuttal.

5. Updates (post-rebuttal)

    After reading the other reviews, as well as the authors' feedback, I would like to raise my score from 5 to 6. The main reason that prevents me from giving a higher rating is the lack of a baseline of the same type, i.e., able to perform continuous changes w.r.t. a `time` variable (e.g., one suggested in the review). Without such a comparison, the paper is just a proof of concept without any evidence that ODE is the best method for the considered tasks.


References
1. Huang et al. Multimodal Unsupervised Image-to-image Translation. ECCV, 2018.
2. Lee et al. DRIT++: Diverse Image-to-Image Translation viaDisentangled Representations. IJCV, 2020.
3. Anokhin et al. High-Resolution Daytime Translation Without Domain Labels. CVPR, 2020.
4. Tancik et al. Fourier Features Let Networks Learn High-Frequency Functions in Low Dimensional Domains. NeurIPS, 2020.
5. Sitzmann et al. Implicit Neural Representations with Periodic Activation Functions. NeurIPS, 2020.
6. Karras et al. Analyzing and Improving the Image Quality of StyleGAN. CVPR, 2020.
7. Pizzati et al. CoMoGAN: continuous model-guided image-to-image translation. CVPR, 2021.

**Time Spent Reviewing:**

4

---

> ### Author Response · Authors · 2021-08-10
> **Thanks for your constructive comments.**
>
>
> **Diversity comparing to BasisGAN**
>
> While BasisGAN is originally proposed to introduce both strong diversity and fidelity to the conditional image generation through modeling the subspace of parameters, the major objective of our method is to model the gradual appearance changes in conditional image generation by modeling a continuous filter subspace.
> While diversity is not the main objective, we show in Section 4.1 that, the successful modeling of the gradual appearance changes (as visualized in each row of Figure 4 and Appendix Figure J) leads to at least comparable diversity with BasisGAN.
> It is true that parameter generation with a multivariate Gaussian seems more reasonable in order to model complex diversity, achieving the best diversity is however not the main focus of our paper. We will leave this as a direction of future efforts.
> We provide in the following anonymous link further visualizations of other datasets:
>
> [Pix2Pix](https://github.com/fornips/rebuttal/blob/master/Pix2Pix.png)
>
> Please note that the images are all generated with continuously sampled time variables to show the smooth appearance change.
>
> ---
>
> **High-resolution experiments**
>
> The proposed continuous atom generator does not exhibit any issues on scalability.
> Unlike datasets like Edge $\rightarrow$ Bag and Edge $\rightarrow$ Shoe we visualized in the paper, high resolution datasets usually have complex structures thus are less obvious to observe the smooth changes w.r.t. the time variables, therefore we did not include high-resolution experiments.
> We provide visualizations of the high-res experiments in the following anonymous link:
>
> [Pix2PixHD](https://github.com/fornips/rebuttal/blob/master/Pix2PixHD.png)
>
> Our method achieves appealing diversity on the high resolution Cityscapes dataset. And we further provide quantitative comparisons on image quality, diversity, running time, memory consuming, and parameters in the following table.
>
> Methods | LPIPS $\uparrow$ | FID $\downarrow$ | Testing speed (s) | Training memory (MB) | Parameter number
> :---:|:---:|:---:|:---:|:---:|:---:
> Pix2PixHD [c] | 0.0 | 48.85 | 0.0299 | 8,145 | 182,546,755
> BasisGAN [39] | 0.168 | 25.12 | 0.0324 | 8,137 | 154,378,051
> Ours | 0.156 | 24.37 | 0.0310 | 8,061 | 167,956,507
>
> We will add this high-resolution discussion to the revision.
>
> ---
>
> **Human faces translation experiment**
>
> For translation tasks on the UTKFace [45] and the CACD2000 [2], the dataset is divided into six different age clusters, where the first cluster was used for the source domain and the remaining five clusters are adopted as the target domain. The results of human faces translation presented in the Appendix, however, show a total of nine generation results that includes interpolation results to illustrate the new interpolation capability from ODE, rather than simply showing the generation results for five different age groups. Thus, evolution between generated images can be seemingly minimal. According to the feedback, we present advanced results on the UTKFace [45] and the CACD2000 [2] and show qualitative comparison with CycleGAN in the following anonymous link:
>
> [HumanFace](https://github.com/fornips/rebuttal/blob/master/FaceTranslation.png)
>
> Note that improved qualitative results are achieved by more epochs of training.
>
> ---
>
> **Line 304 Use of discrete distinct labels**
>
> Conditional image generation on continuous labels is tractable and natural with ODE-based approach, as ODEs are formulated with continuous state variables. We have demonstrated the continuous label case in our paired image-to-image translation experiments, e.g., rotating chairs. In the unpaired image-to-image translation tasks, however, using continuous labels can lead to confusion in training, as the tendency of continuous labels can be violated within unpaired samples. For instance, in the human face translation experiment, since people can have very different levels of aging, it is more natural to have the age group as a class, rather than treating each age as a class. Despite the use of distinct labels, we highlight that it is still possible to capture continuous variation between the generated images via interpolation in the continuous filter subspace.
>
> ---
>
> **Methods with positional embedding**
>
> The positional embedding methods [a,b] provide new directions on modeling an implicit representation through positional embedding.
> The advantages of novel positional embedding can potentially improve the continuous modeling from time variables to atoms.
> However, in our paper, we focus primarily on validating the basic idea of using continuous atom generation to improve conditional image generation with smooth appearance change. Therefore we use the most basic architecture of Neural ODEs in all experiments as shown in the incorporated implementations in supplementary material. The effectiveness of the proposed continuous conditional image generation has been validated on the simplest implementations. We will leave more complex realizations of the continuous functions as a direction of future efforts.
>
> ---
>
> **Comparison to CoMoGAN [d]**
>
> While our method is completely data-driven, the CoMoGAN [d] is a model-guided method, which relies on non-neural physical models to guide the training of continuous image translation. Therefore, the range of the applications of CoMoGAN [d] is highly restricted to the underlying models. For example, while CoMoGAN [d] shows successful results in continuous translation of hue or color, it cannot be deployed in more complex tasks such as face-to-face translation due to the lack of well-defined physical model. We present our qualitative comparison of results of CoMoGAN [d] on Waymo [e] dataset, which is split into source {Dusk/Dawn} and target {Day, Night}, in the following anonymous link:
>
> [Waymo](https://github.com/fornips/rebuttal/blob/master/Waymo.png)
>
> The progressive changes over time in our generated images are produced with interpolation in filter subspace.
>
> ---
>
> [a] Fourier Features Let Networks Learn High Frequency Functions in Low Dimensional Domains, arXiv.
>
> [b] Implicit Neural Representations with Periodic Activation Functions, NeurIPS 2020.
>
> [c] High-Resolution Image Synthesis and Semantic Manipulation with Conditional GANs, CVPR 2018.
>
> [d] CoMoGAN: Continuous Model-Guided Image-to-Image Translation, CVPR 2021.
>
> [e] Scalability in Perception for Autonomous Driving: Waymo Open Dataset, CVPR 2020.

---

> > ### Comment · Reviewer_FG2b · 2021-08-26
> > **Changing the score**
> >
> > I would like to thank the authors for their efforts and thorough feedback. Based on this, I have updated my review and raised the score.

---

### Official Review · Reviewer_aPp5 · 2021-07-16

**Rating:** 6
**Confidence:** 3

**Summary:**

The authors build on previous conditional image generation models, which are based on conditional Generative Adversarial Networks (cGAN). The authors use the observation made by previous work [1, 2], that the kernel $F$ of a convolution layer can be replaced with a low-rank, two-component tensor factorization $F = AD$. $A$ is called the coefficient tensor, and has dimensions [channel_in x channel_out x m]. $D$ has dimensions [m x kernel_width x kernel_height] and represents the $m$ "atoms" of the convolution kernel.

The authors' goal is to devise a cGAN, such that the generator's output is a smooth function of its conditioning input. To this end, the authors propose to model "atoms tensor" $D = D(t)$ using a neural ODE, which turns the atoms into a smooth function of the introduced fictitious time variable $t$. Given some further mild requirements, the authors show that smoothness in the filter atoms induces smoothness in the outputs. Based on this theoretical contribution, they propose novel ways to design cGANs that can be conditioned on images, on some real-valued label or both.

The authors verify the effectiveness of their method on several different tasks, for which they achieve results that are comparable or exceed the state-of-the-art while using much fewer parameters than most comparable methods.

[1] Qiang Qiu, Xiuyuan Cheng, Guillermo Sapiro, et al. DCFNet: Deep neural network with decomposed convolutional filters. ICML 2018

[2] Ze Wang, Xiuyuan Cheng, Guillermo Sapiro, and Qiang Qiu. Stochastic conditional generative networks with basis decomposition. ICLR 2020

**Limitations And Societal Impact:**

There is no discussion about limitations or social impact in the work. Including savings on the parameter count, or inference speed would have been useful. Some explanation and explorations of the theoretical consequences of Theorem 1 would be very useful. Comments on the stability of training and its scalability would be also important.


**Main Review:**

# Strengths
The core idea of the work, i.e. to model the filter atoms using a neural ODE, and then show that the smoothness from the atoms "propagates" to the model's output is a novel and nice, though perhaps somewhat unsurprising contribution.
In my view, the most significant contribution of the work is how this key result is utilised to elegantly allow conditioning on labels whose domain is a subset of the real line.
The methodology seems reasonable, and the authors perform a reasonable extensive empirical verification of their method, and achieve good results.

# Weaknesses
The writing suffers from several mistakes on many levels. It has several spelling and grammatical mistakes, incorrect word usage, incorrect, hard-to-parse and sometimes seemingly superfluous sentences. I also believe that some of the formulae are incorrectly stated.
To give a non-exhaustive list of examples:
- On several occasions, the word "graduate" is used incorrectly to mean "gradual", including in Fig 1.
- Lines 42-43: "First, this method is sensitive to network configurations, and mode collapse, which results in a point estimation to the subspace of parameter, can often be observed." - I cannot parse this sentence.
- Line 74: "[...] with the restriction of condition." - I am not sure what the restriction is supposed to mean here.
- Lines 76-77: "From the perspective of the dynamic of parameter subspace, [...]" - While later sections somewhat clarify what this means, this is extremely confusing at the beginning.
- Lines 99-101: "[16, 27] propose ..." - This sentence belongs to the Related Works section, not the Background section.
- Lines 147-155: The authors seem to explain neural ODEs, even though they have already done so in section 3.1
- Lines 166: I believe that in the definition of the local input norm we the terms in the sum should be $x(u')$ instead of $x(u - u')$, otherwise for every pixel $u$ its local norm would be the local norm around $x(0)$.

Beyond these mistakes, I found the general content layout to be confusing. It took several readings to understand the core idea of "propagating" the Lipschitz continuity from the filter atoms to the outputs, because of the poor explanation.

# Summary
In my opinion, this paper proposes and executes a couple of very nice ideas, which are unfortunately heavily overshadowed by the poor writing. If the authors can heavily improve the writing, I would be happy to recommend acceptance, though I believe this would require major restructuring of the paper.

# After the rebuttal
The authors have addressed my technical concerns, and after some discussion with the other reviewers, I have more confidence that the writing can be improved for the camera-ready version of the paper, hence I raised my score to recommend acceptance.


**Time Spent Reviewing:**

4-5

---

> ### Author Response · Authors · 2021-08-10
> **Thanks for your constructive comments.**
>
> **Writing**
>
> "If the authors can heavily improve the writing, I would be happy to recommend acceptance"
>
> We sincerely apologize for the unpolished text, due to our poor plan to wisely allocate the time between experiments and writing before the due date. We have corrected all the typos, and will for sure further carefully polish and heavily improve the writing in the revision.
>
> ---
>
> **Lines 42-43**
>
> ""First, this method is sensitive to network configurations, and mode collapse, which results in a point estimation to the subspace of parameter, can often be observed." - I cannot parse this sentence."
>
> According to our reimplementation of BasisGAN [39], BasisGAN is sensitive to the network configuration of the basis generator. Following the network configuration presented in [39] exactly can successfully learn the parameter distributions that lead to outputs with diversity and fidelity. However, based on our observations, minor changes to the network configurations, e.g., activation functions and layer sizes, highly likely lead to mode collapse to the parameter distribution, which results in point estimation to the parameter distribution and removes diversity in the output images.
> We will provide more explanations and examples to facilitate interpretations.
>
> ---
>
> **Line 74 Restriction of conditions**
>
> "Line 74: "[...] with the restriction of condition." - I am not sure what the restriction is supposed to mean here."
>
> The restriction here refers specifically to the restriction on using discrete conditions (e.g., class labels). As discussed in [7], models designed for discrete conditions cannot be directly trained with continuous conditions.
>
> ---
>
> **Lines 147-155 Discussion leads to ODE**
>
> "The authors seem to explain neural ODEs, even though they have already done so in section 3.1"
>
> While the content at Lines 147-155 seems overlapping with Section 3.1, especially in terms of equations, the objectives are essentially different.
> We briefly reviewed Neural ODE at the beginning of Section 3 in order to introduce contextual knowledge of our method. In Lines 147-155, we started with assuming all the samples of parameters (filter atoms) jointly reside in a continuous space, and then the movements within this space can be formally formulated as $\boldsymbol{\mathcal{D}}(t+\epsilon) = \boldsymbol{\mathcal{D}}(t) + \epsilon \cdot d\boldsymbol{\mathcal{D}}(t) / dt$, which then naturally leads to a Neural ODE modeling of the underlying dynamic. In summary, Section 3.1 was included to provide background knowledge of Neural ODE, and the discussion at Lines 147-155 was made to naturally explain the steps to model continuous parameter space with Neural ODE. We will further refine the discussion in Section 3.3 to remove potential redundancy.
>
> ---
>
> **Lines 99-101**
>
> "Lines 99-101: "[16, 27] propose ..." - This sentence belongs to the Related Works section, not the Background section."
>
> We raised a few examples on the applications of Neural ODE here in order to facilitate better understanding of the practical advantages of modeling continuous dynamics using Neural ODE, especially for those who are not familiar with Neural ODE. We will reorganize the presentation to prevent misunderstandings. Thank you for your suggestion.
>
> ---
>
> **Line 166**
>
> "I believe that in the definition of the local input norm we the terms in the sum should be $x(u^\prime)$ instead of $x(u - u^\prime)$"
>
> As we defined $N_u \subset \mathcal{U}$ as a local Euclidean grid centered at $u$ in Line 167, $u$ here denotes the coordinates of $x$, therefore $x(u - u^\prime)$ refers to a neighbourhood location near $x$. We will refine the definition here to prevent ambiguities.
>
> ---
>
> **Line 76-77**
>
> "Lines 76-77: "From the perspective of the dynamic of parameter subspace, [...]" - While later sections somewhat clarify what this means, this is extremely confusing at the beginning."
>
> Image generation with continuous conditions only attracts research attention recently, with previous method [7] focuses primarily on solving it with tailored training method. Our method provides a new perspective of modeling the continuous appearance changes through modeling the dynamic of filter subspace, which works well with standard empirical risk minimization training as we show extensively in the experiments. We will further clarify, at the beginning of the paper, our approach of embedding continuous dynamics to the filter subspace in the revision.
>
> ---
>
> **Propagation from continuous parameter subspace to generated images**
>
> To allow image generation with continuous conditions, we adopt Neural ODE to model the dynamic in filter subspace of CNNs with guaranteed continuity. We then show in Theorem 1 that, under our decomposition scheme, the continuity of filter subspace can be propagated to the network outputs, and achieve image generation with smooth appearance changes. Thank you for your insightful suggestion, and we will further clarify this in the abstract.

---

> > ### Comment · Reviewer_aPp5 · 2021-08-31
> > **Response to Rebuttal**
> >
> > I thank the authors for their rebuttal. After also reading the other reviews and the authors' responses, I am generally impressed. I have increased confidence that the writing can be improved for the camera-ready version, hence I now recommend acceptance.

---

### Official Review · Reviewer_Nnbn · 2021-07-18

**Rating:** 7
**Confidence:** 4

**Summary:**

This paper designs continuous filter atoms to enable gradual changes for image generation, by using the ordinary differential equation. This is a clever strategy that decomposes convolutional filters with atoms, and it is significant for the high-dimensional filters. Specifically, those continuous filter atoms are helpful to obtain smooth propagated image generation.  Various experiments on image to image translation and conditioned image generation verify the performances of the introduced filter atoms. Besides, the proposed filter atoms allow users to manipulate images in a new way that controls the integration interval.

**Limitations And Societal Impact:**

- The experimental setting can be improved. The quantitative evaluation is limited. The user study and ablation experiments are missing.

-Paper writing needs to be improved. For example:
- L.039, "stochasiticity" --> "stochasticity"
- L.063, "varying appearance" --> "varying appearances"
- L.066, "without heavy supervisions" --> "without heavy supervision"
- L.075, "has attract attention" --> "has attracted attention"
- L.189, "atoms along the learned continuous" --> "atoms along with the learned continuous"
- L.248, " get the desirable target distribution" --> " get the desired target distribution"
- L.275, "this smooth transitions" --> "these smooth transitions"


**Main Review:**

The idea of using continuous filter atoms from an ODE atom generator is very interesting and meaningful. I believe that it is a very effective and useful method for image generation. The experiments on conditional image generation show that filter atoms cover a wide range of gradually varying appearance and accurate correspondence to the input condition.  Therefore, it allows flexible appearance manipulation without supervision. Also, it can generate image appearance arbitrarily conditioned on continuous labels.

**Time Spent Reviewing:**

72

---

> ### Author Response · Authors · 2021-08-10
> **Thanks for your supportive comments.**
>
> Thank you for your support on the idea and the presentation. We will fix all the issues you pointed in the revision. Please let us know if you have any further comments.
>
> ---
>
> **Experiment settings**
>
> "The experimental setting can be improved."
>
> We organize most of the experiments based on the related methods in the literature to enable fair comparisons with other methods. All evaluation metrics are adopted according to the common practice, and explained in Appendix Section B.2. Most importantly, as a plug-and-play method that works directly by replacing standard convolutional layers, our method does not introduce any auxiliary training objective or regularizations to the network training, therefore no exhaustive tuning of hyperparameters or ablation studies are required. We will further clarify this in the revision.
>
> ---
>
> **Writing**
>
> We have corrected all the spotted typos and will further polish the writing in the revision.

---

### Official Review · Reviewer_t6CV · 2021-07-20

**Rating:** 7
**Confidence:** 2

**Summary:**

The paper describes a novel approach of decomposing convolutional filters into a set of filter atoms, which are modeled by neural ODEs. The authors show how the continuity allows generating gradually changing images.


**Ethical Concerns:**

None.

**Limitations And Societal Impact:**

Weaknesses/Questions
Table 4 doesn’t have the best performing models highlighted in bold.
Typos:
1. Line 39: stochasiticity
1. Line 330: graduate -> gradual


**Main Review:**

Defining the parameters via a composition of filter atoms and implementing these filter atoms via neural ODEs seems like a good idea, introducing continuity without extreme computational costs. The paper is a bit out of my domain of expertise, so I can’t comment much on novelty and evaluation.

Strengths
1. The paper is a pleasure to read. It seems very well organized.
1. The model allows efficient training by leveraging a low-dimensional subspace of filter atoms.
1. Gradual changes of generated images can be learned without the need for auxiliary modules or extra supervision.


**Time Spent Reviewing:**

2

---

> ### Author Response · Authors · 2021-08-10
> **Thanks for your supportive feedbacks.**
>
> Thanks for the support for the idea and the presentation of our paper. We will correct all the typos and mark all the best performance in bold in the revision.
>
> Meanwhile, let us know if you have any further concerns regarding our paper during the discussion period.

---

### Author Response · Authors · 2021-08-10
**We thank all reviewers for their constructive comments.**

All comments have been addressed individually.
We included requested visualizations and quantitative results at the anonymous links provided in each response.

---

### Author Response · Authors · 2021-08-21
**Discussion**

Dear reviewers,

As the ending date of the discussion period is approaching, we look forward to hearing from you for any further discussion. If you have any further concerns, please let us know in this discussion period. We will do our best to response to any follow-up feedback.

We sincerely appreciate your hard work on reviewing our paper.

Thank you!

Authors

---

### Decision · Program_Chairs · 2021-09-27

**Decision:**

Accept (Spotlight)

**Comment:**

This paper proposes using continuous filter atoms (based on Neural ODE) for image generation.  These continuous filter atoms can enable smoother image synthesis compared to convolutional filters. The paper has received positive reviews. Many reviewers find the idea interesting and the results encouraging. The core idea is novel. There are concerns regarding the writing and baselines. The rebuttal addressed most of the concerns. The AC agreed with the reviewers’ consensus and recommended accepting the paper. The authors are encouraged to improve their writing according to the reviewers' comments.